# Technical Note: Temporal Disaggregation of Spatial Rainfall Fields with Generative Adversarial Networks

Sebastian Scher[1,2] and Stefanie Peßenteiner[3]

[1]Know-Center GmbH, Graz, Austria
[2]Stockholm University, Department of Meteorology and Bolin Centre for Climate Research, Stockholm, Sweden
[3]University of Graz, Department of Geography and Regional Science, Graz, Austria

**Correspondence:** Sebastian Scher (sebastian.scher@misu.su.se)

**Abstract.** Creating spatially coherent rainfall patterns with high temporal resolution from data with lower temporal resolution is necessary in many geoscientific applications. From a statistical perspective, this presents a high- dimensional, highly under-determined problem. Recent advances in machine learning provide methods for learning such probability distributions. We test the usage of Generative Adversarial Networks (GANs) for estimating the full probability distribution of spatial rainfall patterns with high temporal resolution, conditioned on a field of lower temporal resolution. The GAN is trained on rainfall radar data with hourly resolution. Given a new field of daily precipitation sums, it can sample scenarios of spatiotemporal patterns with sub-daily resolution. While the generated patterns do not perfectly reproduce the statistics of observations, they are visually hardly distinguishable from real patterns. Limitations that we found are that providing additional input (such as geographical information) to the GAN surprisingly lead to worse results, showing that it is not trivial to increase the amount of used input information. Additionally, while in principle the GAN should learn the probability distribution in itself, we still needed expert judgment to determine at which point the training should stop, because longer training leads to worse results.

## 1 Introduction

Precipitation timeseries in sub-daily temporal resolution are required for numerous applications in environmental modeling. Especially in hydrology, with small to medium catchments whose rainfall-runoff response strongly depends on the temporal rainfall distribution, sub-daily precipitation data is necessary to simulate flood peaks accurately. However, in many settings, precipitation sums only over timescales longer than the needed ones exist. Past sub-daily precipitation records are often only available at short record-lengths (e.g., Breinl and Di Baldassarre, 2019; Lewis et al., 2019; Di Baldassarre et al., 2006) and many future climate projections (GCM-RCM outputs) provide 6-hourly or daily precipitation sums (Müller-Thomy and Sikorska-Senoner, 2019; Verfaillie et al., 2017). To deal with this wide absence of sub-daily precipitation data, several procedures to disaggregate precipitation were proposed in recent years. These include multiplicative cascade models (e.g., Förster et al., 2016; Raut et al., 2018; Müller and Haberlandt, 2018), the method of fragments (e.g., Westra et al., 2012; Sharma and Srikan-

than, 2006) and complex stochastic methods based on e.g. the randomized Bartlett–Lewis model (e.g., Koutsoyiannis and Onof, 2001). Burian et al. (2001, 2000) and Kumar et al. (2012) used artificial neural networks (ANNs) to perform rainfall disaggregation. Pui et al. (2012) provide a comparison of different univariate precipitation disaggregation approaches and an overview of the historical development of precipitation disaggregation frameworks can be found in Koutsoyiannis et al. (2003). Many of these methods are carried out on a station-by-station basis (Müller-Thomy and Sikorska-Senoner, 2019), while others also deal with the more challenging problem of temporal disaggregation of whole spatial fields (e.g., Raut et al., 2018).

In this study, we consider the latter, and we deal with the problem as a purely statistical one. For a given 2D ($n_{lat} \times n_{lon}$) field $c$, representing the daily sum of precipitation, we want to generate a corresponding 3D field of sub-daily precipitation ($t_{res} \times n_{lat} \times n_{lon}$) $y_{abs}$. Since this is a highly under-determined problem, it is our goal to model the probability distribution

$$P(y_{abs}|c) \tag{1}$$

The sum of $y_{abs}$ over the $t_{res}$ dimension must equal to $c$, therefore we can introduce the 3D-vector of fractions of the daily sum $y_{frac}$, defined via

$$y_{frac,tij} = y_{abs,tij}/c_{ij} \tag{2}$$

with $t, i, j$ the indices of the $t_{res}/lat/lon$ dimension, and reformulate the problem as

$$P(y_{frac}|c) \tag{3}$$

with the constraint that

$$\sum_t y_{frac,tij} = 1 \tag{4}$$

Thus we want to model the probability distribution of fractions of the daily precipitation sum, given the daily precipitation sum. The data-dimensionality of this problem increases drastically with increasing size of $n_{lat}$ and $n_{lon}$, as the condition $c$ has a dimensionality of $n_{lat} \times n_{lon}$, and the target $y_{frac}$ the even higher dimensionality $n_{lat} \times n_{lon} \times t_{res}$. Here we use $n_{lat} = n_{lon} = 16$ and $t_{res} = 24$ (corresponding to hourly resolution), thus dimensionalities of 256 and 6144, respectively. This makes statistically inferring the probability distribution $P$ in principle very challenging, even given large amounts of training data. One approach to circumvent this would be building statistical models with information about the underlying problems, and then fitting the parameter of these models to the available observations. However, recent advances in machine-learning have made it possible to directly infer high dimensional probability distributions. The most widely used are Generative Adversarial Networks (GANs) (Goodfellow et al., 2014). GANs are a special class of artificial neural networks that have originally been developed for estimating the probability distribution of images, with the goal of sampling (or "generating") images from this

distributions (widely known as "deep fakes"). Especially in their conditional formulation (Mirza and Osindero, 2014) they are potentially very useful for physics-related problems, such as the one considered in this study. GANs are a very active research field in the machine-learning community and their architectures and training methods are constantly improved (e.g., Arjovsky et al., 2017; Gulrajani et al., 2017; Karras et al., 2018). Given the probabilistic nature of many physical problems, and the high-dimensionality of problems especially in Earth-science related fields, they provide an interesting pathway for new applications.

For example, Leinonen et al. (2019) have used a GAN to infer the 2-D vertical structure of clouds, given 1-D observations of lower resolution satellite observations. GANs have also been used in the modeling of complex chaotic systems (e.g., Wu et al., 2020; King et al., 2018) and have been proposed for stochastic parameterization in geophysical models (Gagne II et al., 2019) and weather forecasting (Bihlo, 2020).

In this study we use measurements of precipitation from weather radars. We train the network on the daily sum of the

measurements and the corresponding 1-hourly patterns of precipitation. To our best knowledge, GANs have not yet been used in the context of precipitation disaggregation. With this study we want assess whether GANs can be a useful tool in temporal precipitation disaggregation. Additionally, we want to provide our developed tool (RainDisaggGAN) as a ready-to-use tool to researchers and practitioners who are interested in creating sub-daily data from spatially distributed daily time series. All the software used for this study, as well as the trained GAN are openly available in the accompanying repository.

Note on terminology: In this study, we use the word "distribution" solely for probability distributions. In the hydrological literature, "distribution" is often also used for spatial and temporal patterns of rainfall. To avoid confusion, here we refer to these strictly as "patterns".

## 2  Methods

### 2.1  Data

We use openly available precipitation radar data from the Swedish meteorological service (SMHI). The data is available from 2009 to present. Here we use measurements from 2009 to 2018. The data covers Sweden and parts of the surrounding area (Fig. 1), and has a temporal resolution of 5 minutes.

The radar reflectivities $Z$ (units $\mathrm{dBZ}$) are converted to rainfall $R$ in $\mathrm{mm\,h^{-1}}$ via

$$R = \left( \frac{10^{Z/10}}{200} \right)^{1/1.5} \tag{5}$$

We then compute the daily sums and use them as condition, and the 24 corresponding 1-hourly fractions as target. The spatial resolution is ~$2 \times 2$ km. We use all available $16 \times 16$ (~$32 \times 32$ km) pixel samples (shifted by 16 pixels, so not including overlapping boxes) from the data that have no missing data in any of the pixels at any time of the day, and that satisfy the following condition: at least 20 pixels must exceed $5\,\mathrm{mm\,day^{-1}}$. This is done to exclude days with very little precipitation from the training. The exact thresholds were chosen without specific physical reasons. We repeated our analysis with $3\,\mathrm{mm\,day^{-1}}$

and $7\,\mathrm{mm\,day^{-1}}$, and the results were similar (not shown). For the training period 2009-2016 this results in 177909 samples,

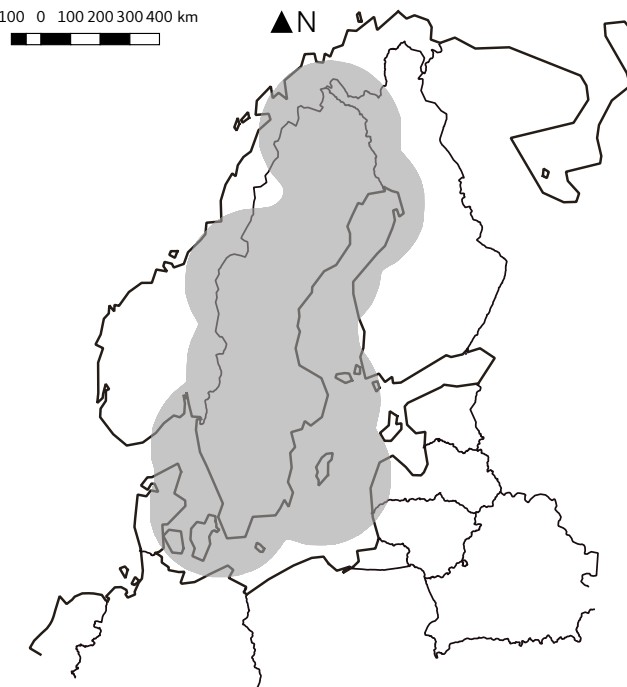

**Figure 1.** Domain of the used SMHI radar data covering most parts of Sweden.

and for the test period 2017-2018 in 59122 samples. We do not differentiate between different precipitation types (e.g. snow, hail) and for readability use "rainfall" and "precipitation" as synonyms. We have to note that excluding all days below the mentioned threshold changes the base sample. Therefore all results on precipitation statistics have to be interpreted in the way that they represent only the statistics for "wet days". This is the case both for the observations shown and for the samples
generated by our GAN-method. The latter is evaluated only on days with rainfall sums above the threshold, therefore they are directly comparable to the observed statistics of all days above threshold. When using our method in practice, one would have to decide whether one also uses the GAN for predicting the patterns for days below the threshold (even though such days have not been included in the training) or not.

## 2.2 GAN

We use the GAN type called Wasserstein-GAN (WGAN) (Arjovsky et al., 2017). A WGAN consists - such as all GANs - of two neural network. The generator, which generates "fake" samples, and a discriminator (called "critic" in WGANs) that judges whether a sample is real or not. In our conditional GAN, the generator takes as input a $16 \times 16$ field of daily sums as condition and a vector of random numbers, and generates a $24 \times 16 \times 16$ field of precipitation fractions. The critic takes as input the $16 \times 16$ condition and a $24 \times 16 \times 16$ sample of fractions, and judges whether it is a fake example or not. The generator and
the critic are trained alternately. The critic is trained with a combination of real and fake examples, and "taught" to differentiate

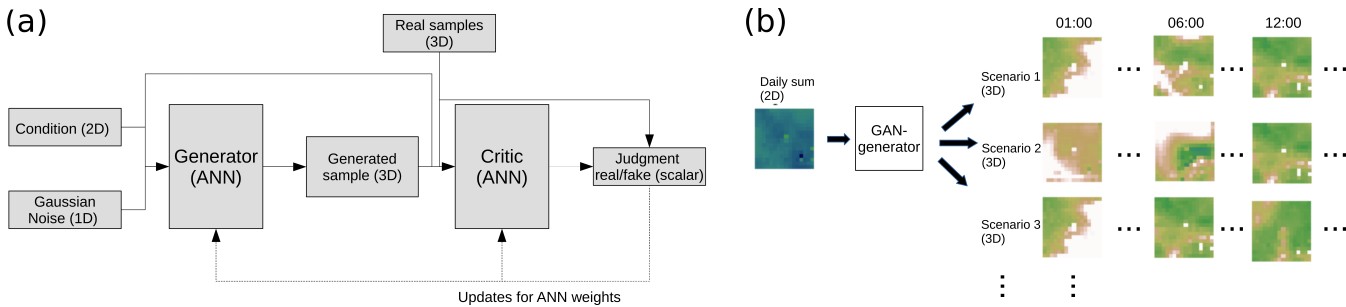

**Figure 2.** (a) principle of a conditional GAN. (b) Sketch of the method.

between them. The generator is then trained to "fool" the critic. The trained generator can then be used to generate fraction scenarios $\hat{y}_{frac}$ from daily sum fields. These can then be converted to precipitation scenarios $\hat{y}_{abs}$ via

$$\hat{y}_{abs,tij} = \hat{y}_{frac,tij} \cdot c_{ij} \qquad (6)$$

A sketch of the principle of a conditional GAN is show in Fig. 2 (a), and our specific approach for generating rainfall
scenarios is is sketched in Fig. 2 (b).

We use a WGAN with gradient penalty (Gulrajani et al., 2017) and pixel normalization (Karras et al., 2018). For details of the training process and to GANs in general we refer to the original papers. Our architecture is based on deep convolutional GANs (DCGAN, Radford et al. (2016)). The input of the generator is a vector of length 100 for the random numbers, and a vector of the flattened $16 \times 16$ condition. This is followed by a fully connected layer of size $256 \times 2 \times 2 \times 3$, three 3D upsampling and
3D convolution layers with increasing dimension and decreasing filter size, each followed by a pixel normalization, and finally a 3D convolution output layer. It would also be possible to use convolutional layers on the input condition before flattening it, even though we have not tested this. Especially for larger domains this might improve the GAN. All layers except the output layer have rectified linear unit (ReLu) activation functions. The output layer uses a softmax layer that does a logistic regression over the $nres$ dimension. With this, the generator automatically satisfies Eq. (4). The critic has a corresponding mirrored
architecture, with 4 strided 3D convolution layers, following the philosophy of using striding instead of downsampling from Gulrajani et al. (2017). Both networks are optimized with the Adam optimizer (Kingma and Ba, 2017) over 50 epochs. After 20 epochs the quality of the generator started to decrease (by visual inspection of samples generated from the train set), therefore we used the saved generator after 20 epochs. Training 20 epochs took 8 hours on a single NVIDIA Tesla V100 GPU. The architecture resulted after some experimentation with different architectures and training methods. The networks were
developed with the Keras (Chollet et al., 2015) and Tensorflow (Martín Abadi et al., 2015) framework. For the details of the architectures, we refer to the Appendix A and the code published together with this paper.

## 2.3 Baseline algorithms

In this section we describe two simple disaggregation methods that are used as baselines in this study.

### 2.3.1 Random Fraction from Training

As simplest baseline, we select random fractions $\boldsymbol{y}_{frac}$ from the training data, and use these fractions together with the daily sum from the testing set as prediction. This is, per definition, unconditional on the daily sum.

### 2.3.2 RainFARM

As further baseline algorithm we use an adaption of the RainFARM method from Rebora et al. (2006). RainFARM is an algorithm for spatiotemporal downscaling of short-term precipitation forecasts. It estimates the slopes $\alpha$ and $\beta$ of the spatial

and temporal power spectra of the low-resolution (low resolution both in time and in space) field. These slopes are assumed constant also for scales smaller than the smallest scales in the low resolution fields. With $\alpha$ and $\beta$, a random power spectrum corresponding to these slopes is generated, with higher spatial and temporal resolution. From this spectrum, a spatial precipitation field is created and scaled to have the same values as the low resolution field on the scales that are covered by the low resolution field.

Here we adopt these ideas to our problem. In contrast to RainFARM, we do not use spatial downscaling, only temporal. Further, since we actually have data with higher time resolution available in the training set, we do not estimate the spectral slopes from the starting field and then extrapolate them, but we estimate them from the hourly resolution data in the training set. Estimating the temporal slope from the starting field would not be possible anyway, because our starting field has only a single timestep (the 24 hour sum).

The temporal slope $\beta$ is estimated the following way: we take a random selection of 500 daily 16x16 fields with hourly resolution. For each day, a FFT is performed over the time-dimension, and the absolute power spectrum for each grid point and each sample is computed. Now all gridpoints and samples are put together in a single linear regression equation, in which the logarithm of the power is regressed against the logarithm of the corresponding wavenumber

$$log\left(P\right) = -\beta \cdot log\left(\omega\right) + c$$

The same is done for the spatial slope $\alpha$. Here, anisotropy for the two spatial dimensions is assumed

$$log\left(P\right) = -\alpha \cdot log\left(\sqrt{k_x^2 + k_y^2}\right) + c$$

Now following the RainFARM algorithm, we generate a Fourier spectrum with the spectral slopes $\alpha$ and $\beta$. In our case it ranges over all spatial scales in a 16x16 pixel part of our data, and from $2\pi/24h$ to $2\pi/1h$ in angular frequency. We start with uniform random distributed phases $\phi\left(k_x, k_y, \omega\right)$, and compute the random power spectrum $\hat{g}$ as

$$\hat{g}\left(k_x, k_y, \omega\right) = e^{-i\phi\left(k_x, k_y, \omega\right)} \sqrt{\left(k_x^2 + k_y^2\right)^{-\alpha/2} \omega^{-\beta}} \tag{7}$$

which subsequently is scaled to unit variance

$$\hat{g}_{scaled} = \hat{g}/std\left(\hat{g}\right) \tag{8}$$

The (not yet correctly scaled) precipitation field $\tilde{r}$ is obtained via inverse FFT and exponentiation

$$\tilde{r} = exp\left(real\left(ifft\left(\hat{g}_{scaled}\right)\right)\right) \tag{9}$$

The actual precipitation field is now computed via scaling $\tilde{r}$, such that the daily sum at each gridpoint corresponds to the input daily sum for each sample

$$r\left(x,y,t\right) = p_{obs}\left(x,y\right)/r\left(x,y,t\right) \tag{10}$$

For each daily sum, an infinite number of different realizations $r$ can be obtained via starting from different random phases $\phi\left(k_x,k_y,\omega\right)$.

## 2.4 Additional Inputs

In our main architecture, we use only fields of daily rainfall sums as input. This is the minimum possible architecture, and therefore also the most generic one for different applications. As extension, we also use two alternative architectures that provide additional inputs to the network. In the first alternate architecture, we input the day-of-the-year for each sample as additional input. Since the day-of-the-year $doy$ is a circular variable (1 is as close to 2 as it is to 365), it is converted to 2 variables $d_1, d_2$ via

$$d_1 = sin\left(2\pi\frac{doy}{365}\right), d_2 = cos\left(2\pi\frac{doy}{365}\right) \tag{11}$$

and these two variables are expanded (repeated) to have the same size as the precipitation input field,and added as additional channels to the input condition. Leap-days are treated as 1st of January.

In the second alternate architecture we use the longitude of the sample as additional input (normalized to $[0,1]$). While Sweden has a much larger North-South than East-West extent, the typical precipitation patterns are more dependent on longitude than latitude, because of the large contrast in orography (mountains in the West, flat in the middle, coast in the East).

## 2.5 Validation

Validation of a trained GAN is a complex and difficult topic. The high-dimensional probability distribution we want to infer is per definition not known (otherwise we would not need the GAN), therefore we cannot directly validate it. This is an inherent problem in generative modelling. We therefore will compare certain statistical properties of the generated samples with the same statistical properties of the real data. We also want to point out that while our method is conditional, it should not be confused with a standard supervised learning method. In the latter, one would assume that there is one "correct" target and train the predictions on this target. In our problem we also have one observed target sub-daily precipitation pattern, but we assume that there are infinitely many possible target patterns for each condition. Our approach is thus much closer to purely generative modelling (learning only the general distribution of rainfall patterns, not conditioned on anything) than to supervised learning

(including probabilistic methods in supervised learning). Still, if interpreted with care, it is possible to use probabilistic forecast evaluation scores. Here we use the widely used Continuous Ranked Probability Score (CRPS; Hersbach (2000)):

$$CRPS(F,y) = \int_x (F(x) - H(x-y))^2 dx \tag{12}$$

with the cumulative distribution function (CDF) $F(x)$ of the forecast distribution, the true value $y$ and the Heaviside step function $H(x)$,

The computation is done with the "properscoring" Python library, which estimates the CRPS via the empirical cumulative distribution function build up by the different scenarios (for each event in the test set, 10000 scenarios are made with the GAN). The CRPS is computed on a forecast-by-forecast basis. For each forecast, we compute the CRPS for each gridpoint and each hour of the day separately, and then average over all gridpoints and hours. Finally, the CRPS is averaged over all forecasts in the test set.

Additionally, we compare the distributions of the generated events with the distributions of the observations. This does not give any information on how well the generated events for a given daily sum correspond to the real daily distribution of that particular day, but it gives information on whether the overall distribution is correct. For this we compute Empirical Cumulative Distribution Functions (ECDFs) and logarithmic spectral distances. The computation of the spectral distances is done in the spatial domain. The hours of the day are treated as individual samples. For each spatial field, the discrete radial absolute Fourier power spectrum $p_i(\omega)$ is computed. Then, for each combination of spatial fields and corresponding spectra $p_1,p_2$, the logarithmic spectral distance $d$ is computed as

$$d(p_1,p_2) = \sqrt{\sum_i^{N_\omega} (10 \cdot log(p1(\omega_i)/p2(\omega_i)))^2 / \omega_i} \tag{13}$$

Additionally, the spectral distance between different methods/observations is computed as well. Here, the distances between all combinations of samples from method1 (GAN) and samples from method2 (RainFARM) are computed.

## 3  Results

Figures 3 and 4 show examples of generated rainfall distributions for two randomly chosen daily sum conditions from a randomly chosen location. For each case, 10 hourly patterns are generated with the same daily sum condition from the test dataset. The figures are to be read as follows: the first row in panel (a) shows the per-gridpoint fraction of observed precipitation every 3 hours, the following rows present the scenarios generated by the GAN (the same figure but in hourly resolution can be found in the Appendix B1,B2). Panel (b) shows the same information, but then scaled by the daily precipitation sum at each gridpoint. In the example of Figure 3 precipitation occurred relatively evenly distributed over the whole day. As expected, due to the many possible patterns that can be associated with a single daily sum, there is a lot of variation in the GAN generated

patterns. In row 3, for example, precipitation is concentrated in the first half of the day, whereas in the next-to-last row, it is concentrated at the end of the day. Patterns generated by RainFARM are displayed in Figure 5. RainFARM generates hourly precipitation in regions where the original daily precipitation field is nonzero, and rainfall is distributed quite evenly over the whole day. This matches the observed situation in Fig. 5 panel (a) well, but doesn't conform with the short term rainfall observed in panel (b), which is better captured by the GAN scenarios (Fig. 4). More examples are included in the accompanying data and code repository. Except from boundary problems at the outermost pixels, the patterns seem to be indistinguishable by eye. In applications were the boundary problem would be an issue, one could use a larger domain and then remove the boundary. Figure 6 shows area means of precipitation per hour. Each panel shows the real pattern for one condition (in black), and 100 patterns generated from the same condition (in green). While it is important that individual samples look reasonable, it is also crucial that the generated sample follow the same distributions as the real pattern. Albeit it is impossible to check whether the GAN recreates the full inter-dependent probability distribution (as we use the GAN to solve this problem in lack of a better method), we can at least check whether the typical sub-daily distribution is captured by the GAN. In the real data, the fractions are not equally distributed over the day, meaning that some times of the day often have higher fractions of the daily sum than others. For this, we randomly select 10000 samples from the test data, and generate a single generator example for each. Then we analyze the daily cycle of the 10000 real patterns and the 10000 generated ones. The result is shown in Fig. 7 (a) (Appendix, Fig. C1 (a) including outliers). When looking at the fractions, the generated distribution seems in general to reasonably follow the real distribution. There are, however, some deviations, mainly an underestimation of the daily cycle. When it comes to the daily cycle of precipitation corresponding to these fractions, the generator does a worse job. Here the daily cycle is even more under-estimated, thus the generator has too little dependency of precipitation on the hour of day. As additional validation, panel (b) in Fig. 7 shows cumulative distribution functions of the observed and generated hourly precipitation patterns, for the same data as the daily-cycle analysis. Shown are both the distribution of the area means, and of point-observations. The plots are capped to exclude very low precipitation amounts. The full plots are shown in the Appendix Fig. C1 (b). In general the distribution of the generated patterns follows the distribution of the observations well. However, they generate too many hourly events with precipitation amounts around $1 \, \mathrm{mm \, h^{-1}}$, and on gridpoint level, the GAN extents to higher maximum precipitation amounts. At very low precipitation amounts (Appendix, Fig. C1 (b)) the distributions seem to be very different. This might be caused by the fact that the softmax output of the network cannot generate values of exactly zero for the fraction, and thus the precipitation cannot be exactly zero neither, except when the daysum is zero. Here, however, one has to consider that such extremely small precipitation amounts are usually of no importance. Additionally, due to the way the data is stored, the radar data cannot go down to zero, but has a minimum slightly above $10^{-4} \, \mathrm{mm \, h^{-1}}$.

Figure 8 shows the distribution of spectral distances of the observations, of the generated patterns and the distribution of spectral distances between observed and generated patterns, all computed on 500 randomly selected samples from the test set. The spectral differences are very similar for both methods and for the observations. Notable is that the observations have a slightly wider right-hand tail than the generated ones, especially compared to RainFARM.

Next, we check whether the GAN actually learns to use the condition input. It could be that the GAN only learns the general distribution of precipitation patterns, without connecting it to the daily sum at all. This could in principle partly be answered by

**Table 1.** CRPS scores of GAN and baseline mehods for 10000 test samples.

| Method | GAN | RainFARM | Random |
|---|---|---|---|
| CRPS [mm/hour] | 0.254 | 0.285 | 0.267 |

the green lines in Fig. 6, however this is difficult to do by eye, and it would also be hard to differentiate between the influence
of the condition, and the influence of the randomness of the noise used as input for the generator. Therefore, we also generated
10 examples for each real one, using the same noise for all 4 panels. Thus generated sample 1 uses the same noise for all
conditions, and sample 2 uses the same (different from sample 1) noise for all conditions and so on. The result is shown in
the 10 colored lines in Fig. 6. The patterns generated for different conditions are similar, but not identical. For example, the
blue line has a distinct peak between 15 and 20 h only in panel (a), and the peak of the yellow line between 1-5 h is slightly
different in all panels. This means that dependent on the condition, different daily fractions are produced. As additional test
on the influence of the condition, we randomly select two conditions, sample 1000 patterns from each condition (using the
same 1000 noise vectors for each condition), and then compute the distribution for each hour of the day, similarly to Fig. 7.
The result for two distinctly different conditions is shown in Fig. 9 (Fig. D1 with outliers). As can be seen, the distributions
are not the same for both conditions. At 10 of the 24 hours of the day, the distributions are significantly different ($p<0.05$ with
two-sample Kolmogorov-Smirnov test). For conditions that are very similar, there is no significant difference at any hour of
the day (not shown). This confirms the result from above that the GAN has at least to some extent learned to use the condition.
Verifying the conditional relationships is difficult to impossible: the high dimension of the condition would make any type of
binning or grouping either in very low sample size for each group, or in groups whose conditions are different only in some of
the dimensions, and therefore a verification is not attempted here.

Finally, we compute the CRPS of the generated patterns. The result is shown in Table 1. The GAN has a slightly lower
CRPS than RainFARM and Random, indicating that it has a higher probabilistic skill. While the difference is small, it is highly
significant. This was tested with the one-sample t-test against the null hypothesis that the mean of the differences is zero ($p=0$,
down to machine precision), and with bootstrapping ($p<0.01$). A one-sample test was chosen because all three methods were
tested on exactly the same test samples, and the scores are therefore not independent.

## 3.1 Architectures with additional inputs

When training our first alternative architecture with day-of-year as additional input, the training did not succeed, and only
resulted in very unrealistic generated patterns (not shown). This might be caused by the fact that with day of the year as input,
the network could be susceptible to overfitting, especially since we are excluding non-rain days from the training data. The
second alternative architecture - with longitude as additional input - was slightly more successful. The results for the longitude
architecture are shown in the Appendix (Fig. E1). Shown is the daily cycle of 10000 randomly selected observations, and
scenarios generated from these. The results are presented for all samples in the Eastern and in the Western half of the domain
separately. The daily cycle of the generated samples is not very realistic, in neither of the domains. In general, even though less

dramatic than the day-of-year input, the additional longitude input seems to have disturbed the training of the network, and in fact made it worse than the base architecture without any additional inputs.

## 4    Discussion and conclusion

In this study we used a Generative Adversarial Network (GAN) to generate possible scenarios of hourly precipitation fields, conditioned on a field of daily precipitation sums. The network was trained on eight years (2009-2016) of hourly observations of Swedish precipitation radar data and the corresponding fields of daily precipitation sum. The trained network can generate reasonable looking hourly scenarios, and thus seems to be able to approximate the probability distribution of the spatiotemporal rainfall patterns. By eye, the generated patterns are nearly indistinguishable from the real patterns. Additionally, the spectral distances of the generated patterns are similar to the observed ones. We showed that the network does not simply learn a general distribution of precipitation patterns, but it also is able to use the conditional daily sum field to some extent. It thus learns a dependency of the probability distribution of rainfall patterns on the daily sum. This result is supported by the GAN CRPS being slightly better but in the same order of magnitude as CRPS of randomly selected and RainFARM generated patterns. We were, however, not able to find a reasonable way to verify this inferred dependency on the daily sum, and its quality hence remains unverified for now. Close inspection of the statistics of many generated samples showed partial agreement but also some deviation from the real statistics, pointing to potential limitations of the method, at least in its current implementation. Adding additional information (longitude and day-of-the-year) to our GAN-architecture was not successful. Not only was the GAN unable to learn the influence of the additional parameters on the probability distribution of the patterns, it also did worse on the general probability distribution. This shows that it is not possible to simply add additional information to an architecture that works without additional inputs. The "black-box" nature of GANs makes it hard to even speculate about a possible reason for this. A further question left open, is the suitability of the presented approach for larger domain sizes. The $16 \times 16$ (~$32 \times 32$ km) pixel approach was chosen having a small catchment in the Alps in mind, which we plan to study in more detail. Recent tests with a $64 \times 64$ pixel domain revealed a decreasing performance. However, such tests are computationally very expensive, and we suspect that our training might have been too short (too few training epochs). Notwithstanding this, this topic needs further inspection. With our GAN-architecture, it is not possible to ensure spatial and temporal continuity when applying the GAN to regions or timeframes that are neighbors in time or space. This could be remedied with the approach presented in Leinonen et al. (2020). Furthermore, as a cautionary remark, we point out that the approach presented in this paper only makes sense if the training data comes from the same climatic conditions as the input data, and the trained GAN should not (or only after careful evaluation) be transferred to other regions. This is because spatial rainfall distributions differ between different (climatic) regions. Finally, expert judgment is still necessary in deciding when to stop the training of the GAN, since after some time the quality of the generated patterns deteriorated. This can potentially be a sever limitation in practice. Detailed study and development will therefore be necessary to further improve the method, and also to make it less dependent on expert judgment.

We conclude that with our current knowledge, it is in principle possible to use GANs in the context of spatial precipitation disaggregation, however only with care and in addition with expert judgement. We hope that this study serves as a starting ground for the hydrological community to work further on assessing the potential of GANs for precipitation disaggregation.

This study was mainly intended as a proof of concept, in order to assess whether it is principally possible to use GANs for temporarily disaggregating spatial rainfall patterns. Whether the method also proofs useful in rainfall-runoff modeling will be assessed in a follow-up study. This runoff modeling could include future climate scenarios. In such a setting it has to be noted that our method – as most other methods – makes a stationarity assumption, meaning that it assumes that the probability distribution of rainfall patterns is always the same (except for the dependency on the daily rainfall sum). In a future (warmer) climate, however, the typical patterns might be different.

When the problem with additional inputs that we encountered are eventually solved, it would be interesting to also test inputting other meteorological variables such as temperature, windspeed or air pressure. These might contain information on the current weather pattern, which itself can have an impact on the possible sub-daily precipitation patterns. This might also be a way to – at least partly – deal with the problem of non-stationarity in future climate scenarios mentioned above. Additionally, our method could be combined with other machine-learning methods. For example, an unsupervised classification scheme could be used prior to training the GAN. With this, the events could be categorized into a number of different classes, and then an individual GAN could be trained for each class. Alternatively, the class-label could be added as input to the GAN. The latter would take the burden of implicitly classifying the events from the GAN.

It would also be of interest to modify the loss-function used for the training of the networks and include constraints on the statistics of the data (for example the reproduction of the daily cycle), following the ideas of Wu et al. (2020). This might eliminate the problems of deviation from the real statistics mentioned earlier. Another option would be to step back from the purely data-driven approach, and try to include physical constraints directly in the GAN.

Variational autoencoders (Kingma and Welling, 2014), which are another type of neural network that can be used to infer high-dimensional (potentially conditional) probability distributions, might also be an attractive alternative to the GAN presented here.

Finally, from a scientific point of view it would be a very appealing attempt using techniques from the emerging field of explainable AI (Samek et al., 2017; Adadi and Berrada, 2018) for the challenging task of using the trained GAN for inferring knowledge about the underlying physical processes.

*Code and data availability.* The SMHI radar data can be freely obtained from http://opendata-download-radar.smhi.se/. The software developed for this study, as well as the trained generator, are available in SSs github repository at https://github.com/sipposip/pr-disagg-radar-gan. Additionally, on final publication, the repository will be archived at Zenodo under the reserved doi 10.5281/zenodo.3733065.

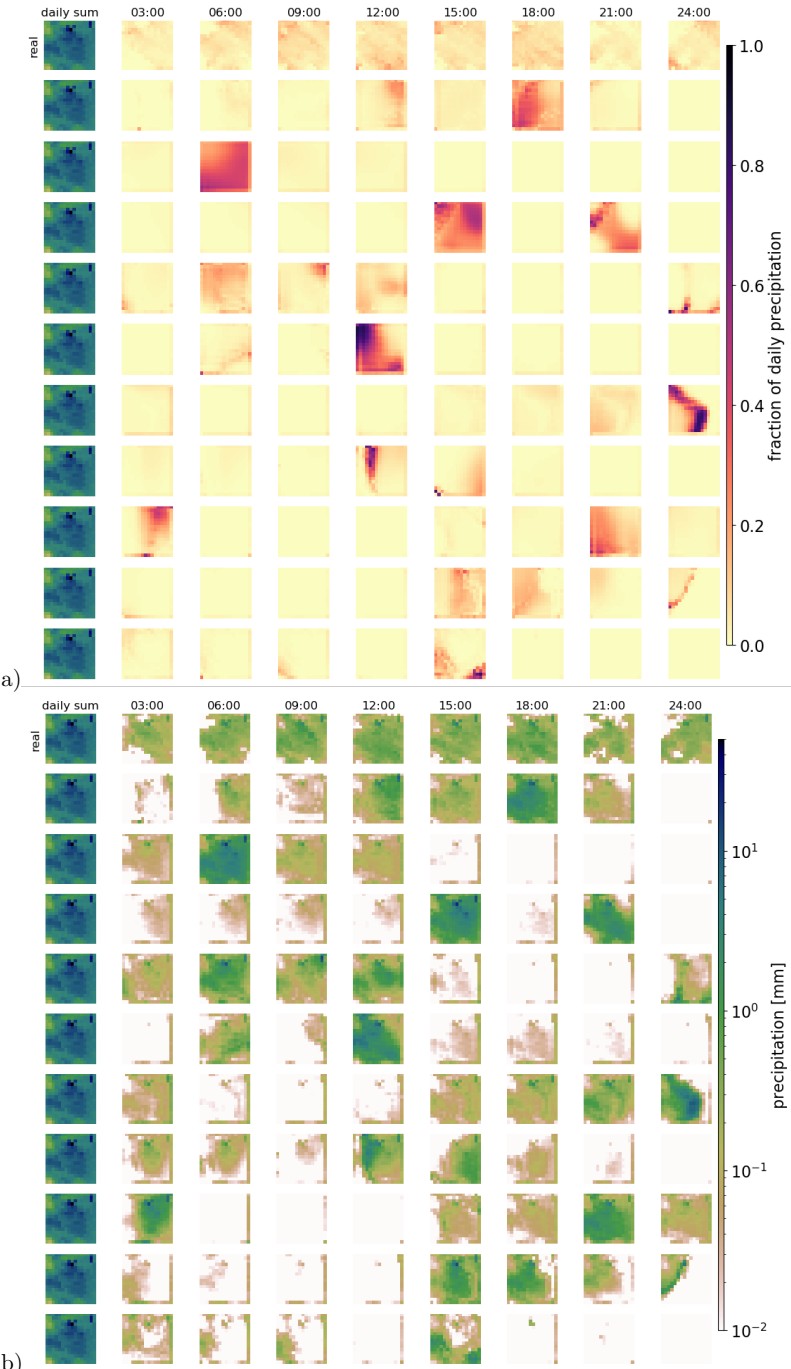

**Figure 3.** Real and generated examples of the fraction of hourly precipitation patterns for one daily precipitation sum, and the hourly precipitation itself. (a) shows the generated fractions, and (b) the corresponding hourly precipitation patterns. The leftmost column of each panel displays the daily sum precipitation field used as condition. The remaining columns show the values for every third hour. The first row presents the observed distribution over the day. The remaining rows show examples generated bye the GAN.

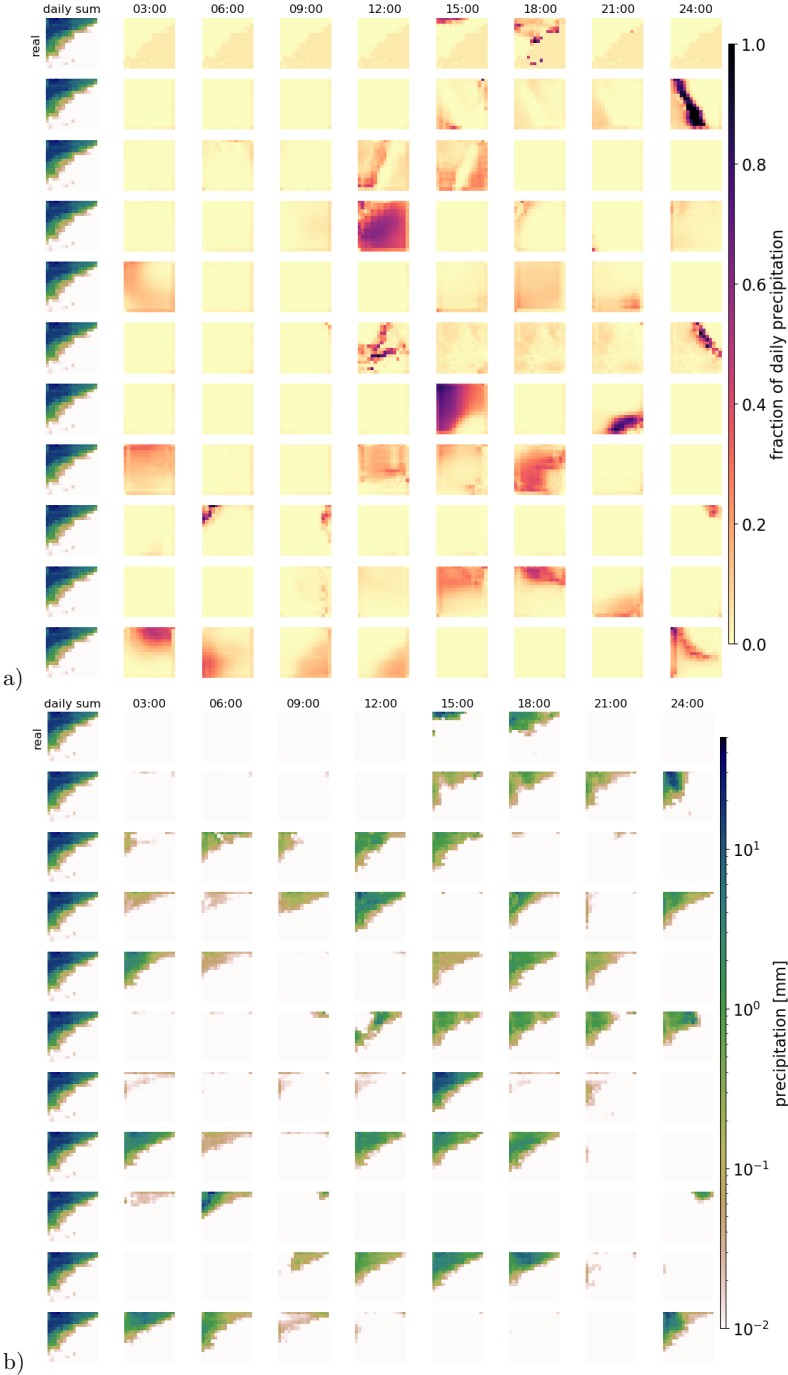

**Figure 4.** As fig. 3, but for a different daily precipitation sum. Note that for very low precipitation amounts, fractions (panel a) might still be high.

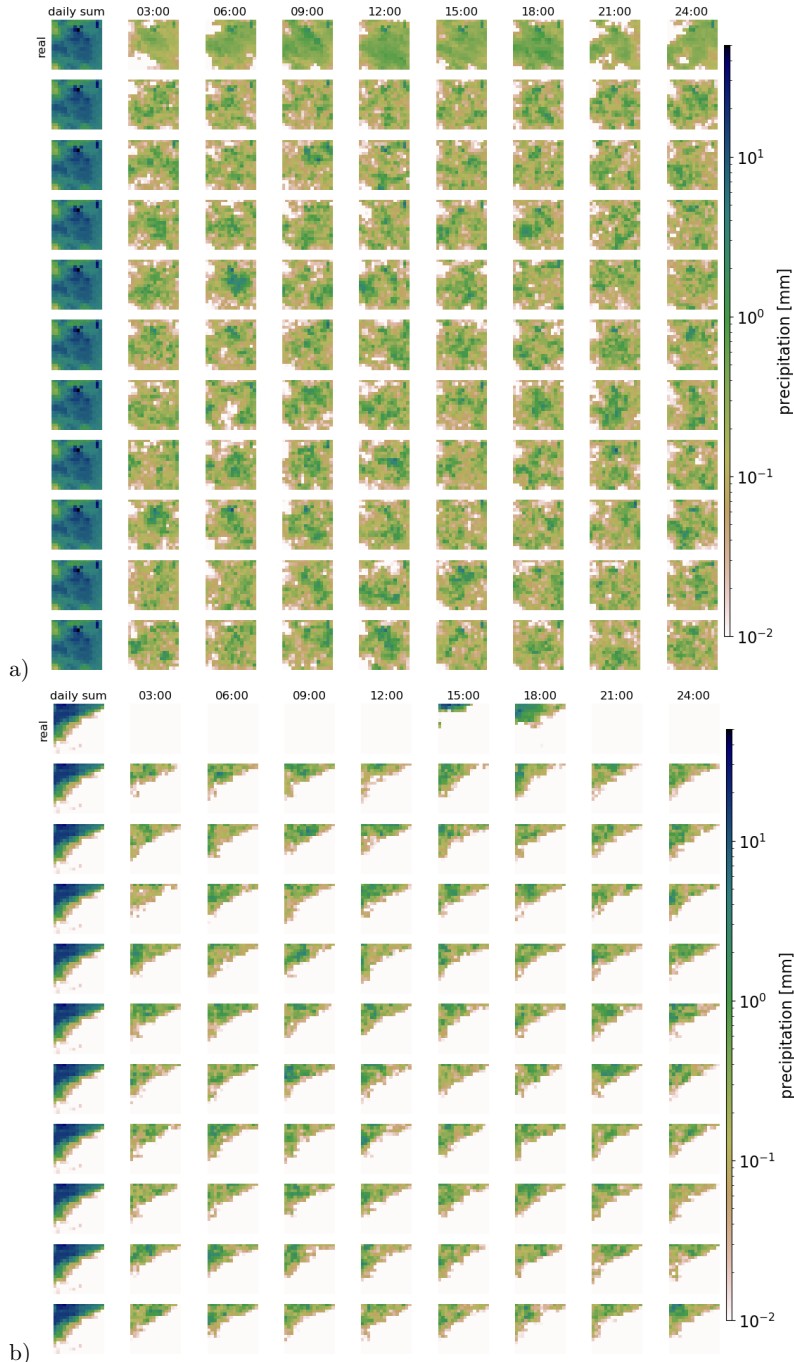

**Figure 5.** Patterns generated with the RainFARM baseline algorithm, for the same days as in 3 and 4.

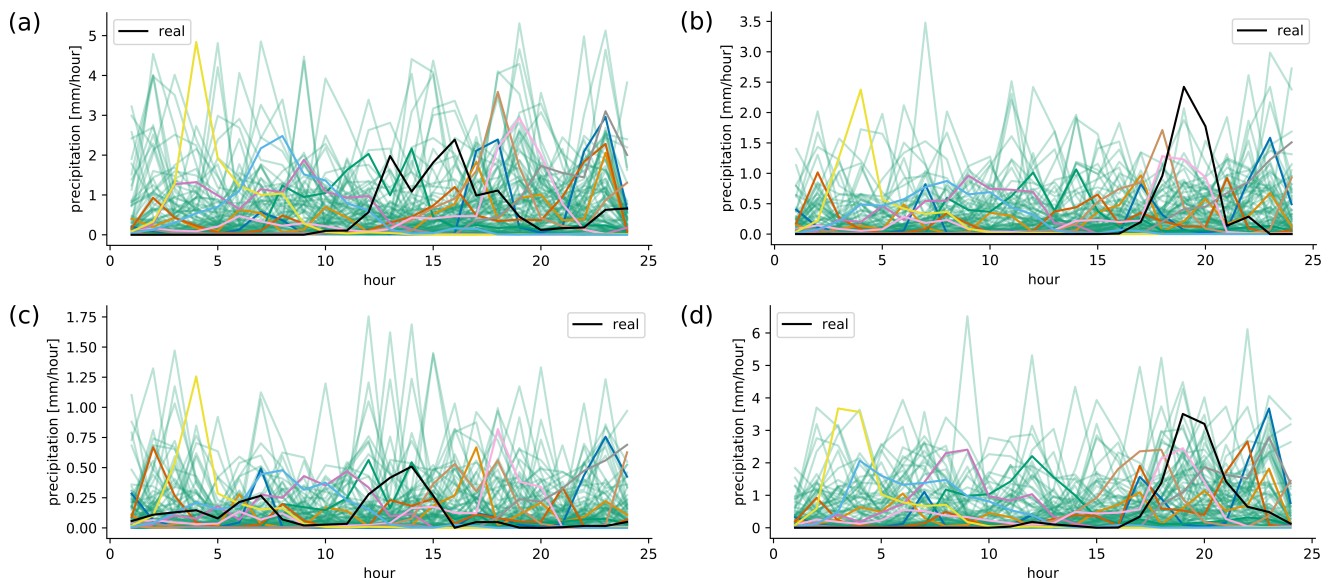

**Figure 6.** Examples of area averaged precipitation scenarios over a single day. The black line shows the observed precipitation, the green lines show 100 generated ones. The colored lines show 10 generated ones, were each color uses exactly the same noise in all 4 plots.

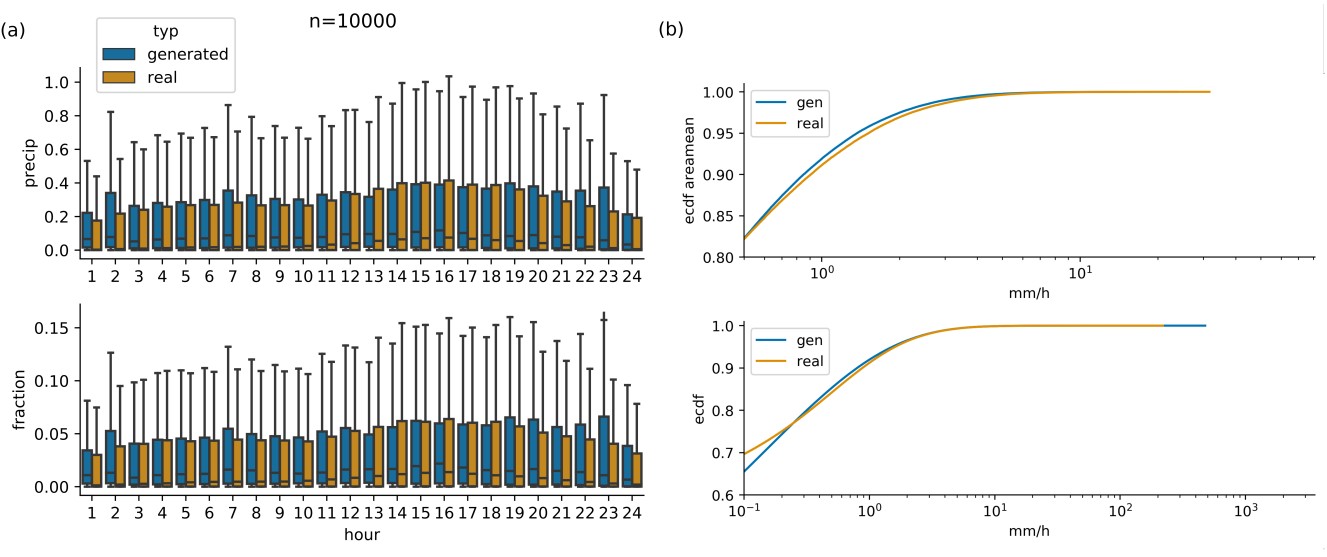

**Figure 7.** (a) Daily cycle of 10000 randomly selected real observations, and scenarios generated by conditioning on exactly the same 10000 daily sums. (b) cumulative distribution functions of generated and observed hourly area mean precipitation (upper panel) and hourly point-level precipitation (lower panel), same data as in (a).

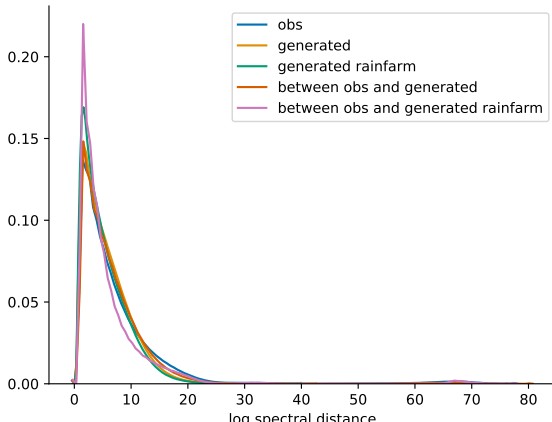

**Figure 8.** Distribution of hourly logarithmic spectral distances ($d$) for the observed patterns (blue), patterns generated with the GAN (yellow), patterns generated with RainFARM (green), and the distances between observed and generated with GAN (red) and between observed and generated with RainFARM (purple), for 500 days (resulting in 12000 hours) from the test set.

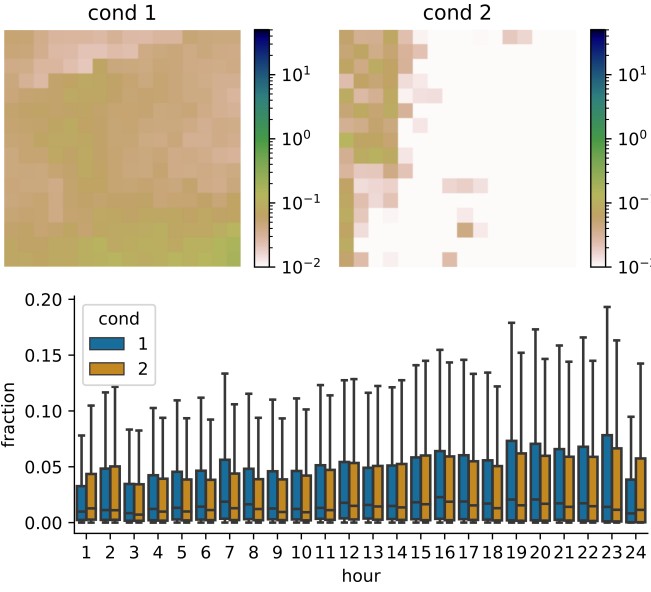

**Figure 9.** Example of daily area mean distributions generated from 2 different daily sum conditions. For each conditions, 1000 scenarios were generated. In all barplots outliers are not shown. The same plots with outliers are shown in the Appendix (D1).

**Appendix A**

The generator consists of the following layers: dense layer with size 3072, upsampling3D, convolution3D (256 channels, kernelsize 2x2x2), pixelnormalization, upsampling3D, convolution3D (128 channels, kernelsize 2x2x2),pixelnormalization, upsampling3D, convolution3D (64 channels, kernelsize 2x2x2), pixelnormalization, convolution3D (1 channel, kernelsize 3x3x3), softmax (along the first dimension).

The critic consists of the following layers:

convolution3D (64 channels), convolution3D (128 channels), convolution3D (256 channels), dense (1 node).

The critic uses dropout regularization (dropout probability 0.25) after each convolution layer. All convolution layers (except the last one in the generator) use a leaky ReLu activation function with $\alpha = 0.2$.

Both the critic and the generator are trained with the the Adam optimizer with parameters $lr = 0.0001$, $\beta_1 = 0$, $\beta_2 = 0.9$ (the values recommend by Gulrajani et al. (2017)).

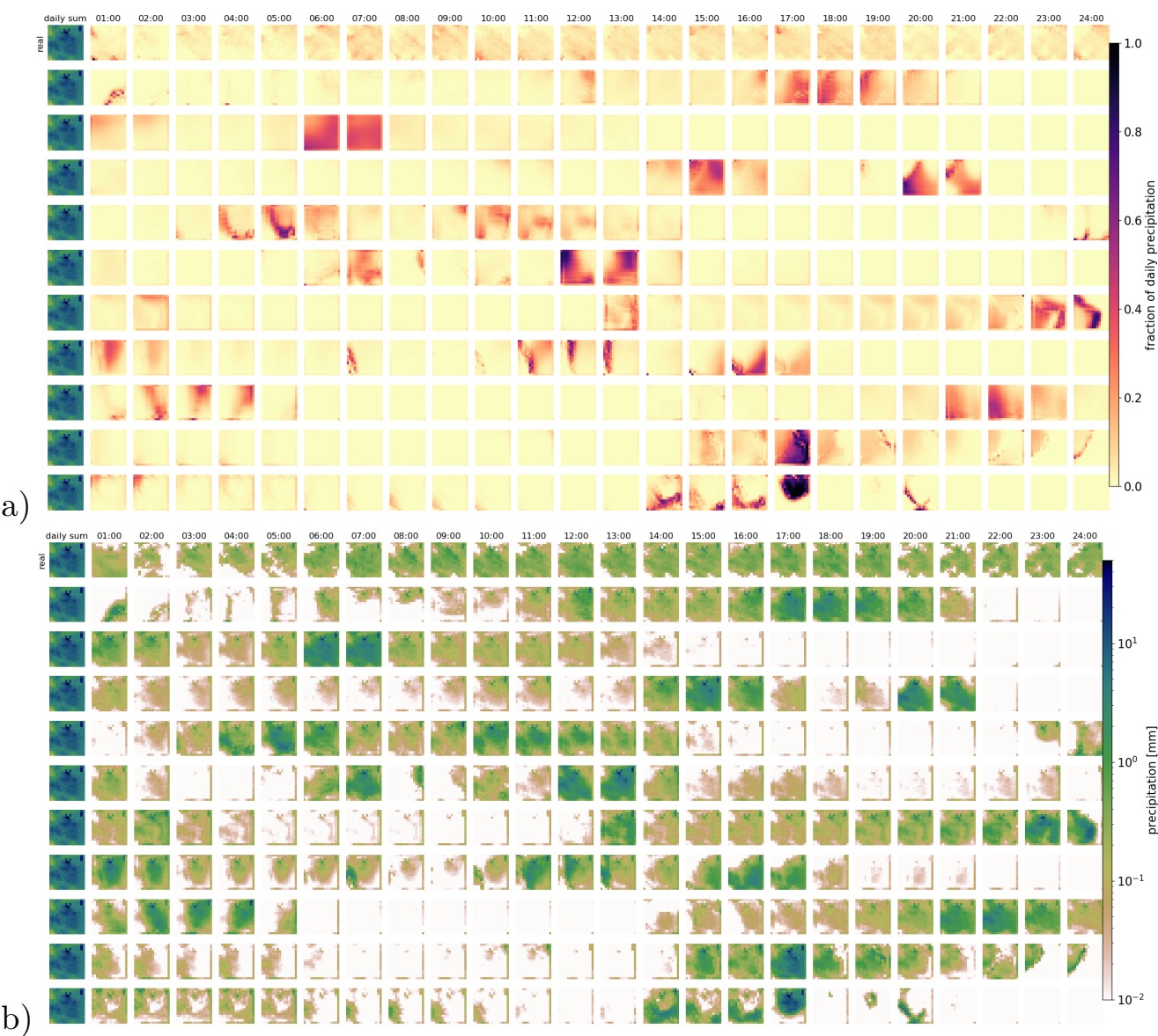

**Figure B1.** As fig. 3, but with all hours shown.

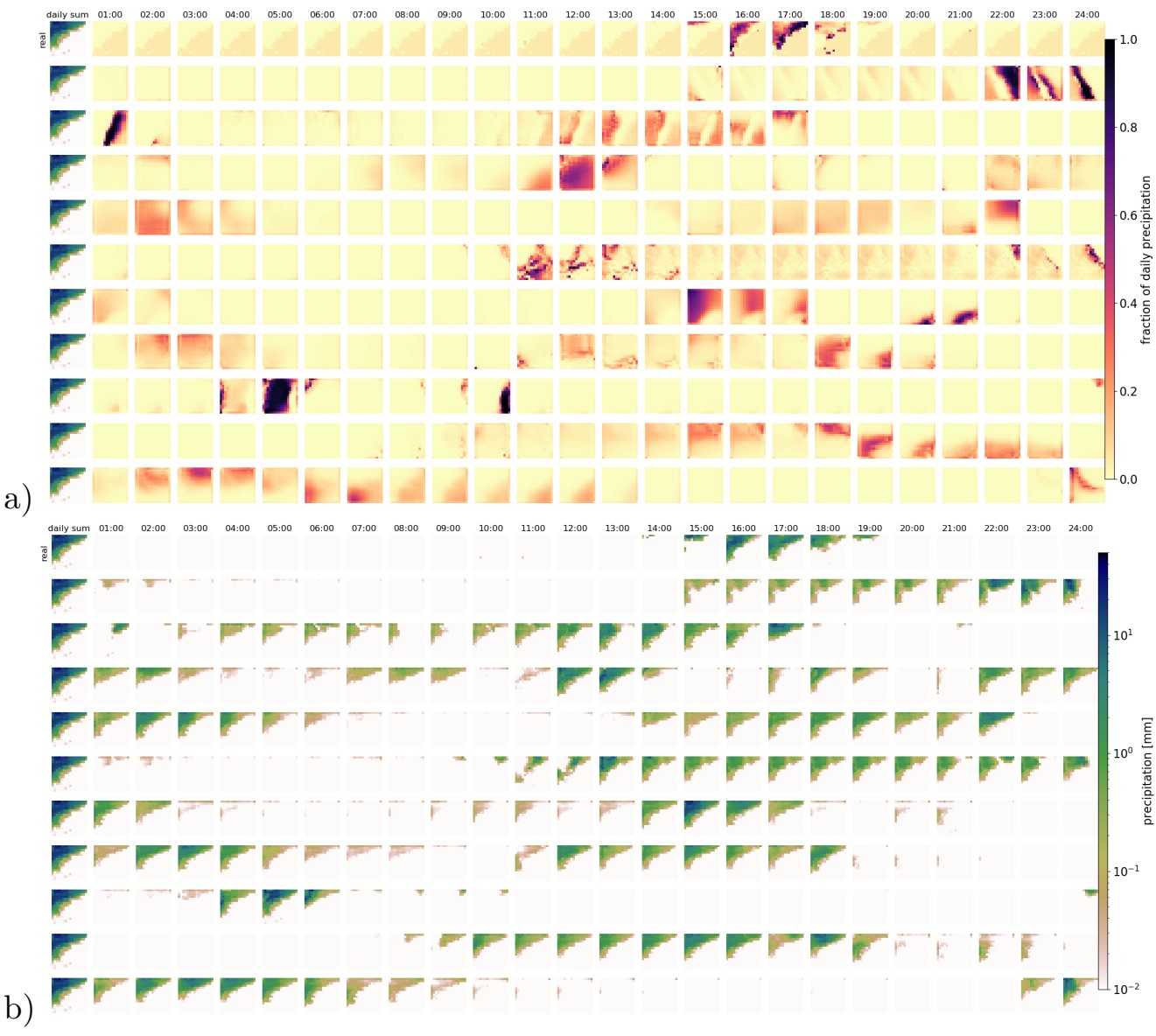

**Figure B2.** As fig. 4, but with all hours shown.

## Appendix C

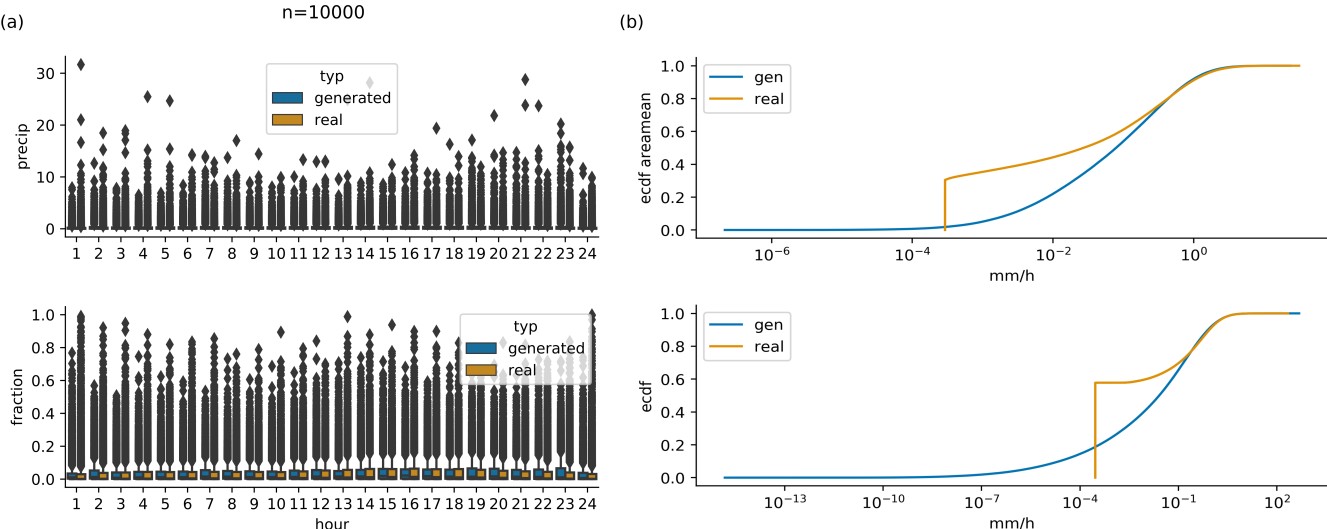

**Figure C1.** (a) Daily cycle of 10000 randomly selected real observations, and scenarios generated by conditioning on exactly the same 10000 daily sums. Same as Fig. **??** (a) but with outliers shown. (b) cumulative distribution functions of generated and observed hourly area mean precipitation (upper panel) and hourly point-level precipitation (lower panel), same data as in (a). Same as Fig. **??** (b), but full range on x-asis.

**Appendix D**

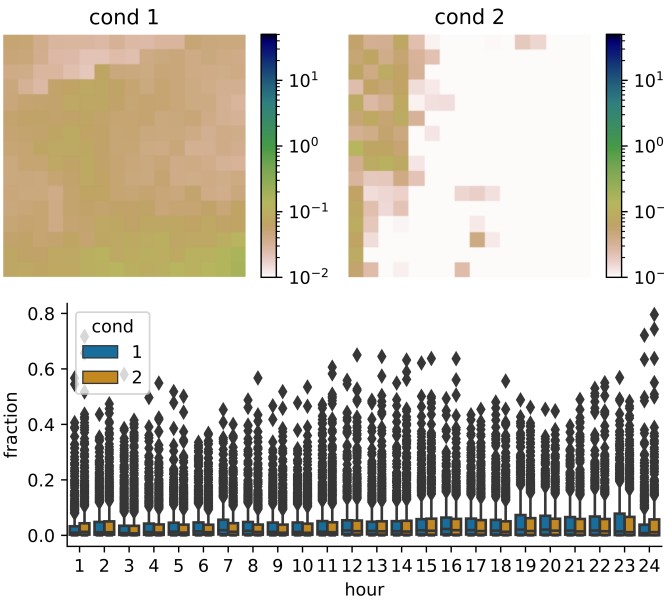

**Figure D1.** Example of daily area mean distributions generated from 2 different daily sum conditions. For each conditions, 1000 scenarios were generated. In all barplots outliers are not shown. The same plots as Fig **??** but with outliers.

## Appendix E

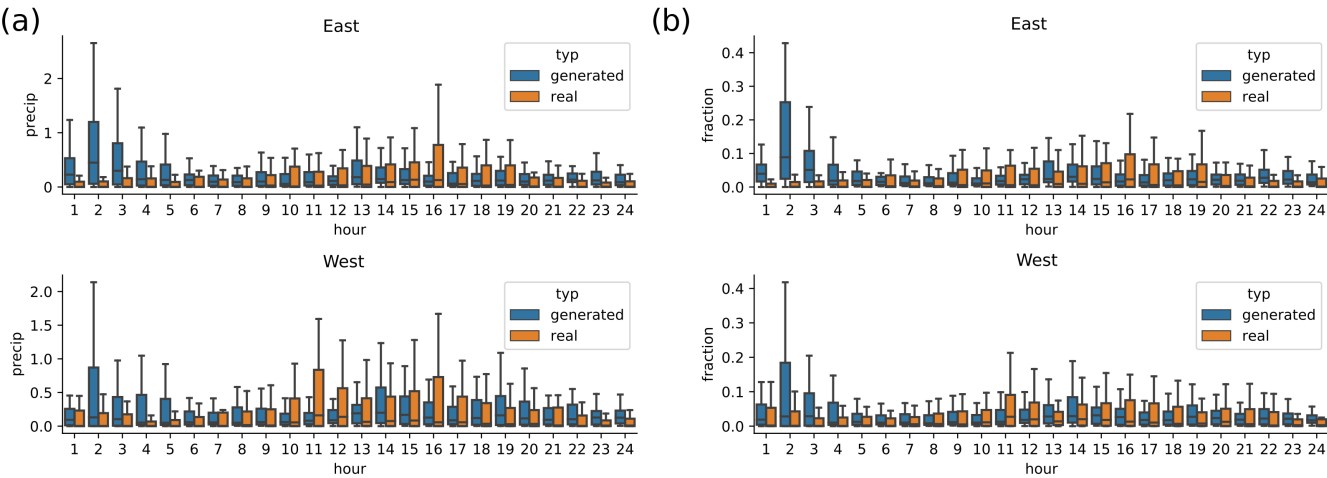

**Figure E1.** Results from the network with longitude as additional input. Daily cycle of 10000 randomly selected real observations, and scenarios generated by conditioning on exactly the same 10000 daily sums, split up into Western and Eastern half of the domain.

*Author contributions.* SP initiated the study. SS developed and implemented the GAN, analyzed the data and drafted the manuscript. Both authors interpreted the results and helped in improving the manuscript.

*Competing interests.* The authors declare that they have no conflict of interest.

*Acknowledgements.* We thank Lea Beusch for interesting discussions. The computations were done on resources provided by the Swedish National Infrastructure for Computing (SNIC) at the High Performance Computing Center North (HPC2N) and National Supercomputer Centre (NSC). The authors acknowledge the Swedish Meteorological and Hydrological Institute (SMHI) for making the radar data freely available. SP has been funded by the Austrian Climate Research Programme (ACRP, 9th call; project RunSed-CC ,grant number KR16AC0K13305).

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
