# Peer review of "Technical Note: Temporal Disaggregation of Spatial Rainfall Fields with Generative Adversarial Networks"

_Hydrology and Earth System Sciences, 2020_

## Referee Comment (RC1) · Jussi Leinonen (Referee) · 14 Dec 2020

*Review by: Jussi Leinonen*

*I'm referring a fair bit to my own work in this review. I do believe these are valid comments and suggestions that could benefit the authors' work, but in the interest of transparency, I'm signing this review with my own name.*

In this manuscript, the authors demonstrate the use of a generative adversarial network on rainfall disaggregation, that is, given a daily total of rainfall as a 2D field, they generate rainfall fields at an hourly resolution that sum up to the daily total at each

point.

The paper is generally clearly written and the methods are presented adequately, although the short-form "Technical Note" format is probably keeping the authors from providing as much explanation as they could have. In any case, appropriate references are provided where the methods are not described in full.

The authors' approach is promising although I feel there are a few shortcomings that should at least be discussed in the text. First, the resolution of the generated fields is very limited, just 16x16 pixels. With the GAN architecture the authors are using, there doesn't seem to be a way to use the trained network to generate rainfall fields of larger extent. If the network is used repeatedly on adjacent 2D tiles of a larger rainfall field, one cannot enforce continuity between the neighboring tiles. Similarly, the generated field is limited to a single day and there is no way to ensure continuity of the rainfall field between the last hour of a given day and the first hour of the next.

In our recent paper (doi:10.1109/TGRS.2020.3032790, or https://arxiv.org/abs/2005. 10374), coauthors and I addressed these two issues by using a fully convolutional architecture to enable the use of the network with different-sized inputs after training, and a recurrent layer to handle the time continuity. Looking at your code in GitHub, I think that the fully convolutional architecture, in particular, would be quite easy to implement and would allow you to generate disaggregated rainfall maps for the entire Sweden with the current training dataset. If you want, I could send you a pull request as an example on how to modify the generator and the critic - let me know!

In any case, I think the paper is publishable with minor revisions as it demonstrates the potential of the method in this application quite effectively and I think this is enough of an incremental advance at this stage, but giving more thought to the abovementioned issues would have made it a better paper. Hopefully these issues can be addressed in later work by the authors.

Specific comments:

Figures: The rainfall maps shown in many figures are really tiny, making these figures quite a strain to the eye to look at. Of course in the PDF one can zoom in but it should be possible to look at the figures in print as well. I'm not sure how to best improve them given that you have 24 frames, but maybe it would be possible to show fewer examples in larger size?

Line 130: Using the day of the year as an input sounds quite susceptible to overfitting as the network can more easily just memorize some input combinations (especially as you're discarding non-raining days). Maybe this is why you're seeing a degraded performance after adding it?

Lines 144-151: This discussion only seems to be about Fig. 4? This is not mentioned here, and also nothing about Fig. 3 is discussed.

Lines 148-149: "In rows 3-4" - minor suggestion, maybe you should number the rows in the figures?

Fig. 3: Where is the prominent feature in the middle of the real precipitation coming from? This seems to be almost stationary throughout the day. Is this real precipitation or a radar artifact? (Also Fig. B1 seems to have a similar feature)

Lines 174-175: Due to the softmax activation, the generator output cannot go to 0 either, so your network also cannot generate zero precipitation (except where the daily total is 0).

Fig. 6a: There seem to be noticeable increases in the generated fractions at hours 2 and 23. I wonder if these could be due to edge artifacts of the convolutional network.

Line 260: Why set $\beta_1$ in Adam to 0? This basically disables this feature of the Adam optimizer. No problem if it works for you, but did you follow some particular implementation?

464, 2020.
Interactive
comment

---

## Short Comment (SC1) · 17 Dec 2020

Dear Jussi,

Thank you for your thorough and constructive review. We will give detailed responses to all your comments once we have received all reviews.

But we already want to thank you for pointing out your recent paper. The approach indeed seems very promising. While we might not have the resources to incorporate the methods in the experiments for our current paper, we will definitely consider using them in future work. Therefore, we would be very thankful if you could indeed show

as with a pull request on how to modify the generator and critic to include the fully convolutional architecture. Obviously, we would acknowledge this in any future work.

best regards,

Sebastian Scher

---

## Referee Comment (RC2) · Anonymous Referee #2 · 15 Jan 2021

Summary

This study presents a methodology based on GANs to disaggregate daily precipitation fields into hourly time steps. The work is based on a large dataset (2009 to 2018) of radar composites from Sweden and a statistical comparison of the simulated and observed distributions over a two-year test set.

The manuscript is well written and the relevance of its content for publication is undisputable. Clearly, the use of GANs to address the temporal disaggregation problem is very interesting. I enjoyed the formulation of the problem, particularly the idea of looking at fractions of the daily precipitation sum in combination of the softmax activation

function on the output layer.

As the authors state, the study represents an important proof of concept that opens many exciting future developments. I am left only with few major concerns that I would like to discuss with the authors before recommending the manuscript for publication.

Major comments

Lines 42-43:

Why 16 km? Is this purely driven by computational limitations? Have you experimented with larger or smaller domains? This might be too small for many potential applications, particularly by considering the typical size of hydrological basins where it would be interesting to test the approach. Please provide some more context to your choice and, if possible, some indications on the sensitivity of the results to the choice of the domain size.

Lines 62-64:

How general is the model that you have trained? Can it be applied to downscale daily sums to a different domain with different climatic conditions? I suggest including some cautionary remarks so to make clear to the reader that an application over a different region and dataset might require training a new model.

Lines 102-104

I would expect the spatial pattern of the conditioning image to play an important role in the daily distribution of precipitation. That is to say, we can expect a certain relationship between spatial and temporal variability . I wonder therefore if by flattening the input image you are making it harder to the GANs to learn such a relationship. Can you comment on this choice? Did you also try to use convolutional layers in your generator so to convert the input image into a vector?

Section 2.2.1

[Figure]

This part raises my main concern, namely the lack of a probabilistic verification. The authors suggest in multiple occasions that this is not possible, but I do not understand why this should be the case. A traditional probabilistic verification approach should be still possible by taking a univariate stance and comparing the N realizations at each point in time and space to the actual observation. As metrics, you may start with the CRPS and PIT histograms. Because of the univariate assumption, you would not assess the accuracy of the GANs in simulating the spatio-temporal structure of precipitation, but it would nevertheless quantify how well the GANs can estimate the underlying conditional probability function.

Related to the above, if possible I would also recommend including a benchmark, so to provide results in terms of improvement with respect to a baseline. I understand that the implementation from the literature of a stochastic disaggregation model for fields might be challenging, but I encourage the authors to still consider it, as in my opinion it would bring much strength to the work.

Figure 3 (also Figs. 4, B1, B2):

The individual images are too small, which makes the visual comparison of real and generated images very hard. Please consider decreasing the number of columns (e.g., plot only 1 image every 3 hours) and the number of rows (plot fewer realizations). Also, but this might be a matter of personal taste, I would discourage the use of the matplotlib's reversed "hot" colormap for precipitation fractions, as the abundance of near-zero fractions produces plots that are mostly white and yellowish and where details are difficult to distinguish.

Lines 217-220:

Although I acknowledge the importance of visual inspection, it may be still interesting to quantify the accuracy of the generated patterns with an objective metric. For images, this can be done by using metrics based on Fourier power spectra, such as the log-spectral distance, by comparing the difference between generated and observed

patterns in the frequency space. A plot of such a metric through the training epochs (learning curve) might be then used as additional evidence to decide when to stop training.

Minor comments

Lines 83-84:

You could also refer to this condition simply by "wet days".

Line 140:

"Figure 3 and 3"

Line 205:

Consider specifying the actual number of years instead of a using "several years".

---

## Author Comment (AC1) · 10 Feb 2021

Dear Jussi,

Thank you for your thorough and constructive review. We appreciate their generally positive evaluation of our manuscript and your valuable suggestions. We here outline how we intend to address the major concerns. A detailed point-by-point response will be included with the submission of our revised manuscript.

We plan to:

- Adapt the mentioned Figures for better readability.

[Figure]

- Include a discussion on the possibility of implementing the fully convolutional architecture suggested and kindly shared by Reviewer #1,

- Clarify all technical details.

---

## Author Comment (AC2) · 10 Feb 2021

Dear Reviewer,

Thank you for your thorough and constructive review. We appreciate the generally positive evaluation of our manuscript and the valuable suggestions, which we plan to incorporate in our revised article in large part. We particularly agree with the main point of a missing benchmark and necessary probabilistic verification. We here outline how we intend to address the major concerns. A detailed point-by-point response will be included with the submission of our revised manuscript.

[Figure]

We plan to:

- Elaborate more on the size of the chosen domain, the sensitivity of results and potential limitations thereof.

- Add cautionary remarks on the generalization of the trained model to different domains in other climatic regions.

- Include a stochastic disaggregation baseline as to provide a benchmark for our results.

- Incorporate additional evaluation metrics such as the CRPS.

- Adapt the mentioned Figures for better readability.
* * *

---

## Author Response (AR1)

Dear Editor,

We want to thank you for obtaining two very constructive reviews, which greatly helped to improve our manuscript. We have now prepared a new version of our manuscript that takes full account of the reviewers comments. As you will see, we have added two baselines, which help to better put the results of our work in context. Additionally, we have significantly expanded the discussion section and improved the figures, based on the points by both reviewers. Please find below point-by-point responses (black font) to all referee comments (blue font).

Best regards

Sebastian Scher and Stefanie Peßenteiner

**Review #1**

1) The authors' approach is promising although I feel there are a few shortcomings that should at least be discussed in the text. First, the resolution of the generated fields is very limited, just 16x16 pixels. With the GAN architecture the authors are using, there doesn't seem to be a way to use the trained network to generate rainfall fields of larger extent. If the network is used repeatedly on adjacent 2D tiles of a larger rainfall field, one cannot enforce continuity between the neighboring tiles. Similarly, the generated field is limited to a single day and there is no way to ensure continuity of the rainfall field between the last hour of a given day and the first hour of the next.
In our recent paper (doi:10.1109/TGRS.2020.3032790, or https://arxiv.org/abs/2005. 10374), coauthors and I addressed these two issues by using a fully convolutional architecture to enable the use of the network with different-sized inputs after training, and a recurrent layer to handle the time continuity. Looking at your code in GitHub, I think that the fully convolutional architecture, in particular, would be quite easy to implement and would allow you to generate disaggregated rainfall maps for the entire Sweden with the current training dataset. If you want, I could send you a pull request as an example on how to modify the generator and the critic - let me know!

Thank you for pointing to your very promising approach and for sharing your convolutional encoder-decoder type implementation on github. While we unfortunately do not have the resources to incorporate these methods in the experiments for our current paper, we will definitely consider using them in future work. We now mention these limitations of our approach in the conclusion section, and refer to your paper as a possible solution.

Line 130: Using the day of the year as an input sounds quite susceptible to overfitting as the network can more easily just memorize some input combinations (especially as you're discarding non-raining days). Maybe this is why you're seeing a degraded performance after adding it?

Thanks for pointing out this potential reason for the degradation, this is now discussed (L262).

Lines 144-151: This discussion only seems to be about Fig. 4? This is not mentioned here, and also nothing about Fig. 3 is discussed.

A discussion on both figures is now included.

Lines 148-149: "In rows 3-4" - minor suggestion, maybe you should number the rows in the figures?

Thanks for this suggestion. The figures now have a smaller number of rows, and we therefore think that adding row numbers is not necessary, and would only cloud the figures unnecessarily.

Fig. 3: Where is the prominent feature in the middle of the real precipitation coming from? This seems to be almost stationary throughout the day. Is this real precipitation or a radar artifact? (Also Fig. B1 seems to have a similar feature)

This might have indeed been an artifact. However, in the course of the revision, we had to rerun our scripts, and due to the fact that we are randomly sampling test dates and regions, figure 3 has changed since it is now based on another testing sample.

Lines 174-175: Due to the softmax activation, the generator output cannot go to 0 either, so your network also cannot generate zero precipitation (except where the daily total is 0).

Thanks for pointing this out, we now mention this in the text (L229).

Fig. 6a: There seem to be noticeable increases in the generated fractions at hours 2 and 23. I wonder if these could be due to edge artifacts of the convolutional network.

We are not sure whether we completely agree that there is a noticeable increase the these two hours. Also please note that each panel is for a single real example, and for example in panel d this effect is clearly not visible. We therefore do not believe that this is due to boundary effects.

Line 260: Why set β 1 in Adam to 0? This basically disables this feature of the Adam optimizer. No problem if it works for you, but did you follow some particular implementation?

Here we followed the recommendation in https://arxiv.org/pdf/1704.00028.pdf. This is now mentioned in the manuscript.

**Review #2**

1) This study presents a methodology based on GANs to disaggregate daily precipitation fields into hourly time steps. The work is based on a large dataset (2009 to 2018) of radar composites from Sweden and a statistical comparison of the simulated and observed distributions over a two-year test set.
The manuscript is well written and the relevance of its content for publication is undisputable. Clearly, the use of GANs to address the temporal disaggregation problem is very interesting. I enjoyed the formulation of the problem, particularly the idea of looking at fractions of the daily precipitation sum in combination of the softmax activation function on the output layer. As the authors state, the study represents an important proof of concept that opens many exciting future developments. I am left only with few major concerns that I would like to discuss with the authors before recommending the manuscript for publication.

Thank you for the positive evaluation of our work and your constructive comments which helped to improve our manuscript a lot. Please find below the answers to your review.

2) Lines 42-43: Why 16 km? Is this purely driven by computational limitations? Have you experimented with larger or smaller domains? This might be too small for many potential applications, particularly by considering the typical size of hydrological basins where it would be interesting to test the approach. Please provide some more context to your choice and, if possible, some indications on the sensitivity of the results to the choice of the domain size.

The domain size 16×16 pixel (~32×32 km) was originally chosen arbitrarily but having in mind a small catchment we plan to study in more detail. Considering your comment we also tested a domain size of 64×64 pixels , where we find a decreasing performance. However, with our computational resources being limited, we suspect that the training was too short (too few training epochs). We now discuss this in the conclusion section. Additionally, reviewer #1 also mentioned an architecture that should allow for the generation of larger domains. This is now also discussed.

3) Lines 62-64: How general is the model that you have trained? Can it be applied to downscale daily sums to a different domain with different climatic conditions? I suggest including some cautionary remarks so to make clear to the reader that an application over a different region and dataset might require training a new model.

This is indeed a very important point. The method should indeed not be used – or only very cautiously – for downscaling in different climatic regions than the one it was trained on. The method learns the typical spatiotempral distributions of rainfall during the day, and this distribution is not the same in different climatic regions. This is now mentioned in the discussion section (L293 onwards)

4) Lines 102-104: I would expect the spatial pattern of the conditioning image to play an important role in the daily distribution of precipitation. That is to say, we can expect a certain relationship between spatial and temporal variability . I wonder therefore if by flattening the input image you are making it harder to the GANs to learn such a relationship. Can you comment on this choice? Did you also try to use convolutional layers in your generator so to convert the input image into a vector?

You raise a very good point. The choice for flattening the input condition instead of using a convolutional architecture was solely based on simplicity. Intuitively we would not assume that for the small domain (16x16) it would make a large difference, but we have not tested it. Especially for larger domains, however, it would probably make a lot if sense to use convolutional layers for the input condition. This is now discussed in the text (L106)

5) Section 2.2.1 This part raises my main concern, namely the lack of a probabilistic verification. The authors suggest in multiple occasions that this is not possible, but I do not understand why this should be the case. A traditional probabilistic verification approach should be still possible by taking a univariate stance and comparing the N realizations at each point in time and space to the actual observation. As metrics, you may start with the CRPS and PIT histograms. Because of the univariate assumption, you would not as- sess the accuracy of the GANs in simulating the spatio-temporal structure of precipita- tion, but it would nevertheless quantify how well the GANs can estimate the underlying conditional probability function.

Related to the above, if possible I would also recommend including a benchmark, so to provide results in terms of improvement with respect to a baseline. I understand that

the implementation from the literature of a stochastic disaggregation model for fields might be challenging, but I encourage the authors to still consider it, as in my opinion it would bring much strength to the work.

We do agree that a major weakness of our original manuscript was a lack of verification. We have now addressed this in two ways: We have 1) added CRPS as probabilistic validation metric and 2) added two baseline methods (random sampling, and a spectral method).

6) Figure 3 (also Figs. 4, B1, B2): The individual images are too small, which makes the visual comparison of real and generated images very hard. Please consider decreasing the number of columns (e.g., plot only 1 image every 3 hours) and the number of rows (plot fewer realizations). Also, but this might be a matter of personal taste, I would discourage the use of the matplotlib's reversed "hot" colormap for precipitation fractions, as the abundance of near-zero fractions produces plots that are mostly white and yellowish and where details are difficult to distinguish.

Thank you for pointing this out. We adapted our figures for better readability.

7) Lines 217-220: Although I acknowledge the importance of visual inspection, it may be still interesting to quantify the accuracy of the generated patterns with an objective metric. For images, this can be done by using metrics based on Fourier power spectra, such as the log-spectral distance, by comparing the difference between generated and observed patterns in the frequency space. A plot of such a metric through the training epochs (learning curve) might be then used as additional evidence to decide when to stop training.

Thanks for this suggestion. We have now included an analysis of power spectra. As to your suggestion of tracking the difference between the spectra in the learning: this is indeed a very interesting idea. Unfortunately, we did neither have the computational nor the time-resources to test this idea.

8) Lines 83-84: You could also refer to this condition simply by "wet days".

We changed this accordingly.

9) Line 140: "Figure 3 and 3"

Changed to Figures 3 and 4.

10) Line 205: Consider specifying the actual number of years instead of a using "several years".

Changed to: The network was trained on eight years (2009-2016) of hourly observations..."

---

## Author Response (AR2)

Dear Editor,

Thank you for accepting our manuscript.
I have now split up the figures as suggested by you. I have also considered the additional reference suggested by the reviewer. The study they pointed out is definitely interesting, and it indeed builds on the RainFARM algorithm that we also use, but it expands it in a way that in my eyes is not relevant for our study (they do a time-extrapolation to get a nowcast). I think that it would be inappropriate to cite it, as it does not relate to our study, and might even confuse readers who seek the background literature of RainFARM, and I have therefore, for now, not included it.

Best regards,

Sebastian Scher